# Learning What and Where to Draw

**Scott Reed**[1],[*]
reedscot@google.com

**Zeynep Akata**[2]
akata@mpi-inf.mpg.de

**Santosh Mohan**[1]
santoshm@umich.edu

**Samuel Tenka**[1]
samtenka@umich.edu

**Bernt Schiele**[2]
schiele@mpi-inf.mpg.de

**Honglak Lee**[1]
honglak@umich.edu

[1]University of Michigan, Ann Arbor, USA
[2]Max Planck Institute for Informatics, Saarbrücken, Germany

## Abstract

Generative Adversarial Networks (GANs) have recently demonstrated the capability to synthesize compelling real-world images, such as room interiors, album covers, manga, faces, birds, and flowers. While existing models can synthesize images based on global constraints such as a class label or caption, they do not provide control over pose or object location. We propose a new model, the Generative Adversarial What-Where Network (GAWWN), that synthesizes images given instructions describing what content to draw in which location. We show high-quality $128 \times 128$ image synthesis on the Caltech-UCSD Birds dataset, conditioned on both informal text descriptions and also object location. Our system exposes control over both the bounding box around the bird and its constituent parts. By modeling the conditional distributions over part locations, our system also enables conditioning on arbitrary subsets of parts (e.g. only the beak and tail), yielding an efficient interface for picking part locations.

## 1 Introduction

Generating realistic images from informal descriptions would have a wide range of applications. Modern computer graphics can already generate remarkably realistic scenes, but it still requires the substantial effort of human designers and developers to bridge the gap between high-level *concepts* and the end product of pixel-level details. Fully automating this creative process is currently out of reach, but deep networks have shown a rapidly-improving ability for controllable image synthesis.

In order for the image-generating system to be useful, it should support high-level control over the contents of the scene to be generated. For example, a user might provide the category of image to be generated, e.g. "bird". In the more general case, the user could provide a textual description like "a yellow bird with a black head".

Compelling image synthesis with this level of control has already been demonstrated using convolutional Generative Adversarial Networks (GANs) [Goodfellow et al., 2014, Radford et al., 2016]. Variational Autoencoders also show some promise for conditional image synthesis, in particular recurrent versions such as DRAW [Gregor et al., 2015, Mansimov et al., 2016]. However, current approaches have so far only used simple conditioning variables such as a class label or a non-localized caption [Reed et al., 2016b], and did not allow for controlling where objects appear in the scene.

To generate more realistic and complex scenes, image synthesis models can benefit from incorporating a notion of localizable objects. The same types of objects can appear in many locations in different scales, poses and configurations. This fact can be exploited by separating the questions of "what"

---

[*]Majority of this work was done while first author was at U. Michigan, but completed while at DeepMind.

and "where" to modify the image at each step of computation. In addition to parameter efficiency, this yields the benefit of more *interpretable* image samples, in the sense that we can track what the network was meant to depict at each location.

For many image datasets, we have not only global annotations such as a class label but also localized annotations, such as bird part keypoints in Caltech-USCD birds (CUB) [Wah et al., 2011] and human joint locations in the MPII Human Pose dataset (MHP) [Andriluka et al., 2014]. For CUB, there are associated text captions, and for MHP we collected a new dataset of 3 captions per image.

Our proposed model learns to perform location- and content-controllable image synthesis on the above datasets. We demonstrate two ways to encode spatial constraints (though there could be many more). First, we show how to condition on the coarse location of a bird by incorporating spatial masking and cropping modules

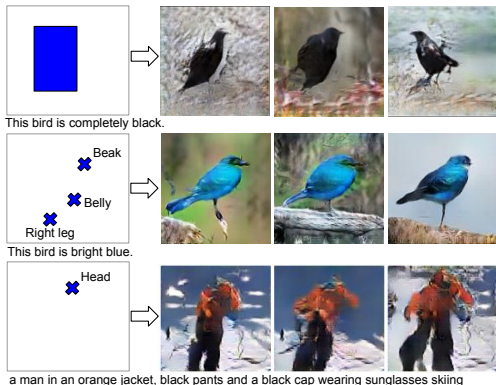

Figure 1: Text-to-image examples. Locations can be specified by keypoint or bounding box.

into a text-conditional GAN, implemented using spatial transformers. Second, we can condition on part locations of birds and humans in the form of a set of normalized (x,y) coordinates, e.g. `beak@(0.23,0.15)`. In the second case, the generator and discriminator use a multiplicative gating mechanism to attend to the relevant part locations.

The main contributions are as follows: (1) a novel architecture for text- and location-controllable image synthesis, yielding more realistic and higher-resolution CUB samples, (2) a text-conditional object part completion model enabling a streamlined user interface for specifying part locations, and (3) exploratory results and a new dataset for pose-conditional text to human image synthesis.

## 2 Related Work

In addition to recognizing patterns within images, deep convolutional networks have shown remarkable capability to *generate* images. Dosovitskiy et al. [2015] trained a deconvolutional network to generate 3D chair renderings conditioned on a set of graphics codes indicating shape, position and lighting. Yang et al. [2015] followed with a recurrent convolutional encoder-decoder that learned to apply incremental 3D rotations to generate sequences of rotated chair and face images. Oh et al. [2015] used a similar approach in order to predict action-conditional future frames of Atari games. Reed et al. [2015] trained a network to generate images that solved visual analogy problems.

The above models were all deterministic (i.e. conventional feed-forward and recurrent neural networks), trained to learn one-to-one mappings from the latent space to pixel space. Other recent works take the approach of learning probabilistic models with variational autoencoders [Kingma and Welling, 2014, Rezende et al., 2014]. Kulkarni et al. [2015] developed a convolutional variational autoencoder in which the latent space was "disentangled" into separate blocks of units corresponding to graphics codes. Gregor et al. [2015] created a recurrent variational autoencoder with attention mechanisms for reading and writing portions of the image canvas at each time step (DRAW).

In addition to VAE-based image generation models, simple and effective Generative Adversarial Networks [Goodfellow et al., 2014] have been increasingly popular. In general, GAN image samples are notable for their relative sharpness compared to samples from the contemporary VAE models. Later, class-conditional GAN [Denton et al., 2015] incorporated a Laplacian pyramid of residual images into the generator network to achieve a significant qualitative improvement. Radford et al. [2016] proposed ways to stabilize deep convolutional GAN training and synthesize compelling images of faces and room interiors.

Spatial Transformer Networks (STN) [Jaderberg et al., 2015] have proven to be an effective visual attention mechanism, and have already been incorporated into the latest deep generative models. Eslami et al. [2016] incorporate STNs into a form of recurrent VAE called Attend, Infer, Repeat (AIR), that uses an image-dependent number of inference steps, learning to generate simple multi-object 2D and 3D scenes. Rezende et al. [2016] build STNs into a DRAW-like recurrent network with impressive sample complexity visual generalization properties.

Larochelle and Murray [2011] proposed the Neural Autoregressive Density Estimator (NADE) to tractably model distributions over image pixels as a product of conditionals. Recently proposed spatial grid-structured recurrent networks [Theis and Bethge, 2015, van den Oord et al., 2016] have shown encouraging image synthesis results. We use GANs in our approach, but the same principle of separating "what" and "where" conditioning variables can be applied to these types of models.

## 3 Preliminaries

### 3.1 Generative Adversarial Networks

Generative adversarial networks (GANs) consist of a generator $G$ and a discriminator $D$ that compete in a two-player minimax game. The discriminator's objective is to correctly classify its inputs as either real or synthetic. The generator's objective is to synthesize images that the discriminator will classsify as real. $D$ and $G$ play the following game with value function $V(D, G)$:

$$\min_G \max_D V(D, G) = \mathbb{E}_{x \sim p_{data}(x)}[\log D(x)] + \mathbb{E}_{x \sim p_z(z)}[\log(1 - D(G(z)))]$$

where $z$ is a noise vector drawn from e.g. a Gaussian or uniform distribution. Goodfellow et al. [2014] showed that this minimax game has a global optimium precisely when $p_g = p_{data}$, and that when $G$ and $D$ have enough capacity, $p_g$ converges to $p_{data}$.

To train a conditional GAN, one can simply provide both the generator and discriminator with the additional input $c$ as in [Denton et al., 2015, Radford et al., 2016] yielding $G(z, c)$ and $D(x, c)$. For an input tuple $(x, c)$ to be intepreted as "real", the image $x$ must not only look realistic but also match its context $c$. In practice $G$ is trained to maximize $\log D(G(z, c))$.

### 3.2 Structured joint embedding of visual descriptions and images

To encode visual content from text descriptions, we use a convolutional and recurrent text encoder to learn a correspondence function between images and text features, following the approach of Reed et al. [2016a] (and closely related to Kiros et al. [2014]). Sentence embeddings are learned by optimizing the following structured loss:

$$\frac{1}{N} \sum_{n=1}^{N} \Delta(y_n, f_v(v_n)) + \Delta(y_n, f_t(t_n)) \tag{1}$$

where $\{(v_n, t_n, y_n), n = 1, ..., N\}$ is the training data set, $\Delta$ is the 0-1 loss, $v_n$ are the images, $t_n$ are the corresponding text descriptions, and $y_n$ are the class labels. $f_v$ and $f_t$ are defined as

$$f_v(v) = \arg\max_{y \in \mathcal{Y}} \mathbb{E}_{t \sim \mathcal{T}(y)}[\phi(v)^T \varphi(t)], \quad f_t(t) = \arg\max_{y \in \mathcal{Y}} \mathbb{E}_{v \sim \mathcal{V}(y)}[\phi(v)^T \varphi(t)] \tag{2}$$

where $\phi$ is the image encoder (e.g. a deep convolutional network), $\varphi$ is the text encoder, $\mathcal{T}(y)$ is the set of text descriptions of class $y$ and likewise $\mathcal{V}(y)$ for images. Intuitively, the text encoder learns to produce a higher compatibility score with images of the correspondong class compared to any other class, and vice-versa. To train the text encoder we minimize a surrogate loss related to Equation 1 (see Akata et al. [2015] for details). We modify the approach of Reed et al. [2016a] in a few ways: using a char-CNN-GRU [Cho et al., 2014] instead of char-CNN-RNN, and estimating the expectations in Equation 2 using the average of 4 sampled captions per image instead of 1.

## 4 Generative Adversarial What-Where Networks (GAWWN)

In the following sections we describe the bounding-box- and keypoint-conditional GAWWN models.

### 4.1 Bounding-box-conditional text-to-image model

Figure 2 shows a sketch of the model, which can be understood by starting from input noise $z \in \mathbb{R}^Z$ and text embedding $t \in \mathbb{R}^T$ (extracted from the caption by pre-trained [2] encoder $\varphi(t)$) and following the arrows. Below we walk through each step.

First, the text embedding (shown in green) is replicated spatially to form a $M \times M \times T$ feature map, and then warped spatially to fit into the normalized bounding box coordinates. The feature map

entries outside the box are all zeros.[3] The diagram shows a single object, but in the case of multiple localized captions, these feature maps are averaged. Then, convolution and pooling operations are applied to reduce the spatial dimension back to $1 \times 1$. Intuitively, this feature vector encodes the coarse spatial structure in the image, and we concatenate this with the noise vector $z$.

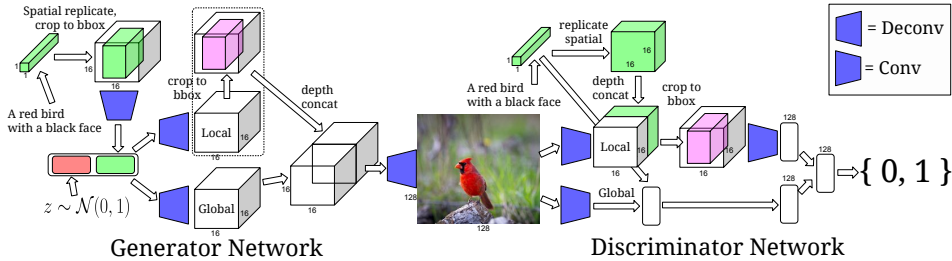

Figure 2: GAWWN with bounding box location control.

In the next stage, the generator branches into local and global processing stages. The global pathway is just a series of stride-2 deconvolutions to increase spatial dimension from $1 \times 1$ to $M \times M$. In the local pathway, upon reaching spatial dimension $M \times M$, a masking operation is applied so that regions outside the object bounding box are set to 0. Finally, the local and global pathways are merged by depth concatenation. A final series of deconvolution layers are used to reach the final spatial dimension. In the final layer we apply a Tanh nonlinearity to constrain the outputs to $[-1, 1]$.

In the discriminator, the text is similarly replicated spatially to form a $M \times M \times T$ tensor. Meanwhile the image is processed in local and global pathways. In the local pathway, the image is fed through stride-2 convolutions down to the $M \times M$ spatial dimension, at which point it is depth-concatenated with the text embedding tensor. The resulting tensor is spatially cropped to within the bounding box coordinates, and further processed convolutionally until the spatial dimension is $1 \times 1$. The global pathway consists simply of convolutions down to a vector, with additive contribution of the orignal text embedding $t$. Finally, the local and global pathway output vectors are combined additively and fed into the final layer producing the scalar discriminator score.

## 4.2   Keypoint-conditional text-to-image model

Figure 3 shows the keypoint-conditional version of the GAWWN, described in detail below.

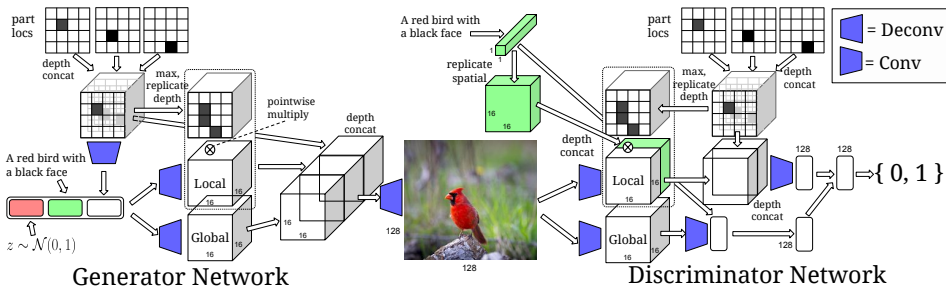

Figure 3: Text and keypoint-conditional GAWWN.. Keypoint grids are shown as $4 \times 4$ for clarity of presentation, but in our experiments we used $16 \times 16$.

The location keypoints are encoded into a $M \times M \times K$ spatial feature map in which the channels correspond to the part; i.e. `head` in channel 1, `left foot` in channel 2, and so on. The keypoint tensor is fed into several stages of the network. First, it is fed through stride-2 convolutions to produce a vector that is concatenated with noise $z$ and text embedding $t$. The resulting vector provides coarse information about content and part locations. Second, the keypoint tensor is flattened into a binary matrix with a 1 indicating presence of *any* part at a particular spatial location, then replicated depth-wise into a tensor of size $M \times M \times H$.

In the local and global pathways, the noise-text-keypoint vector is fed through deconvolutions to produce another $M \times M \times H$ tensor. The local pathway activations are gated by pointwise multiplication with the keypoint tensor of the same size. Finally, the original $M \times M \times K$ keypoint tensor is

depth-concatenated with the local and global tensors, and processed with further deconvolutions to produce the final image. Again a Tanh nonlinearity is applied.

In the discriminator, the text embedding $t$ is fed into two stages. First, it is combined additively with the global pathway that processes the image convolutionally producing a vector output. Second, it is spatially replicated to $M \times M$ and then depth-concatenated with another $M \times M$ feature map in the local pathway. This local tensor is then multiplicatively gated with the binary keypoint mask exactly as in the generator, and the resulting tensor is depth-concatenated with the $M \times M \times T$ keypoints. The local pathway is fed into further stride-2 convolutions to produce a vector, which is then additively combined with the global pathway output vector, and then into the final layer producing the scalar discriminator score.

## 4.3 Conditional keypoint generation model

From a user-experience perspective, it is not optimal to require users to enter every single keypoint of the parts of the object they wish to be drawn (e.g. for birds our model would require 15). Therefore, it would be very useful to have access to all of the conditional distributions of unobserved keypoints given a subset of observed keypoints and the text description. A similar problem occurs in data imputation, e.g. filling in missing records or inpainting image occlusions. However, in our case we want to draw convincing samples rather than just fill in the most likely values.

Conditioned on e.g. only the position of a bird's beak, there could be several very different plausible poses that satisfy the constraint. Therefore, a simple approach such as training a sparse autoencoder over keypoints would not suffice. A DBM [Salakhutdinov and Hinton, 2009] or variational autoencoder [Rezende et al., 2014] could in theory work, but for simplicity we demonstrate the results achieved by applying the same generic GAN framework to this problem.

The basic idea is to use the assignment of each object part as observed (i.e. conditioning variable) or unobserved as a gating mechanism. Denote the keypoints for a single image as $k_i := \{x_i, y_i, v_i\}, i = 1, ..., K$, where $x$ and $y$ indicate the row and column position, respectively, and $v$ is a bit set to 1 if the part is visible and 0 otherwise. If the part is not visible, $x$ and $y$ are also set to 0. Let $\mathbf{k} \in [0, 1]^{K \times 3}$ encode the keypoints into a matrix. Let the conditioning variables (e.g. a beak position specified by the user) be encoded into a vector of switch units $\mathbf{s} \in \{0, 1\}^K$, with the $i$-th entry set to 1 if the $i$-th part is a conditioning variable and 0 otherwise. We can formulate the generator network over keypoints $G_k$, conditioned on text $\mathbf{t}$ and a subset of keypoints $\mathbf{k}, \mathbf{s}$, as follows:

$$G_k(z, \mathbf{t}, \mathbf{k}, \mathbf{s}) := \mathbf{s} \odot \mathbf{k} + (1 - \mathbf{s}) \odot f(z, \mathbf{t}, \mathbf{k}) \quad (3)$$

where $\odot$ denotes pointwise multiplication and $f : \mathbb{R}^{Z+T+3K} \to \mathbb{R}^{3K}$ is an MLP. In practice we concatenated $z$, $\mathbf{t}$ and flattened $\mathbf{k}$ and chose $f$ to be a 3-layer fully-connected network.

The discriminator $D_k$ learns to distinguish real keypoints and text $(\mathbf{k}_{real}, \mathbf{t}_{real})$ from synthetic. In order for $G_k$ to capture all of the conditional distributions over keypoints, during training we randomly sample switch units $\mathbf{s}$ in each mini-batch. Since we would like to usually specify 1 or 2 keypoints, in our experiments we set the "on" probability to 0.1. That is, each of the 15 bird parts only had a $10\%$ chance of acting as a conditioning variable for a given training image.

## 5 Experiments

In this section we describe our experiments on generating images from text descriptions on the Caltech-UCSD Birds (CUB) and MPII Human Pose (MHP) datasets.

CUB [Wah et al., 2011] has 11,788 images of birds belonging to one of 200 different species. We also use the text dataset from Reed et al. [2016a] including 10 single-sentence descriptions per bird image. Each image also includes the bird location via its bounding box, and keypoint (x,y) coordinates for each of 15 bird parts. Since not all parts are visible in each image, the keypoint data also provides an additional bit per part indicating whether the part can be seen.

MHP Andriluka et al. [2014] has 25K images with 410 different common activities. For each image, we collected 3 single-sentence text descriptions using Mechanical Turk. We asked the workers to describe the most distinctive aspects of the person and the activity they are engaged in, e.g. "a man in a yellow shirt preparing to swing a golf club". Each image has potentially multiple sets of (x,y) keypoints for each of the 16 joints. During training we filtered out images with multiple people, and for the remaining 19K images we cropped the image to the person's bounding box.

We encoded the captions using a pre-trained char-CNN-GRU as described in [Reed et al., 2016a]. During training, the 1024-dimensional text embedding for a given image was taken to be the average of four randomly-sampled caption encodings corresponding to that image. Sampling multiple captions per image provides further information required to draw the object. At test time one can average together any number of description embeddings, including a single caption.

For both CUB and MHP, we trained our GAWWN using the ADAM solver with batch size 16 and learning rate 0.0002 (See Alg. 1 in [Reed et al., 2016b] for the conditional GAN training algorithm). The models were trained on all categories and we show samples on a set of held out captions. For the spatial transformer module, we used a Torch implementation provided by Oquab [2016]. Our GAN implementation is loosely based on `dcgan.torch`[4].

In experiments we analyze how accurately the GAWWN samples reflect the text and location constraints. First we control the location of the bird by interpolation via bounding boxes and keypoints. We consider both the case of (1) ground-truth keypoints from the data set, and (2) synthetic keypoints generated by our model, conditioned on the text. Case (2) is advantageous because it requires less effort from a hypothetical user (i.e. entering 15 keypoint locations). We then compare our CUB results to representative samples from the previous work. Finally, we show samples on text- and pose-conditional generation of images of human actions.

## 5.1 Controlling bird location via bounding boxes

We first demonstrate sampling from the text-conditional model while varying the bird location. Since location is specified via bounding box coordinates, we can also control the size and aspect ratio of the bird. This is shown in Figure 4 by interpolating the bounding box coordinates while at the same time fixing the text and noise conditioning variables.

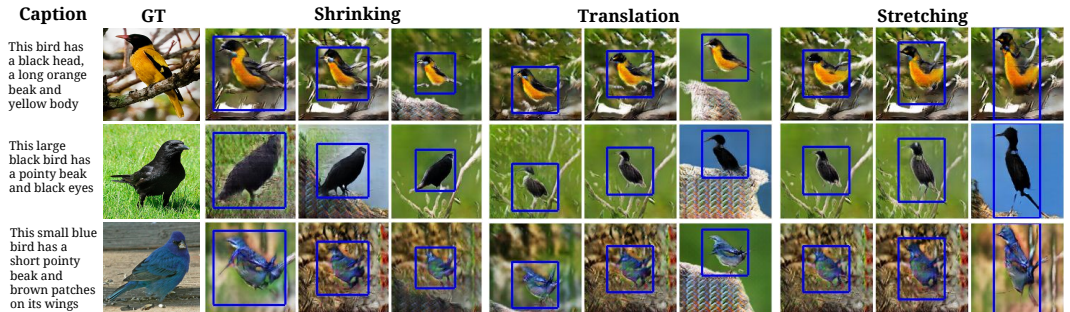

Figure 4: Controlling the bird's position using bounding box coordinates. and previously-unseen text.

With the noise vector $z$ fixed in every set of three frames, the background is usually similar but not perfectly invariant. Interestingly, as the bounding box coordinates are changed, the direction the bird faces does not change. This suggests that the model learns to use the the noise distribution to capture some aspects of the background and also non-controllable aspects of "where" such as direction.

## 5.2 Controlling individual part locations via keypoints

In this section we study the case of text-conditional image generation with keypoints fixed to the ground-truth. This can give a sense of the performance upper bound for the text to image pipeline, because synthetic keypoints can be no more realistic than the ground-truth. We take a real image and its keypoint annotations from the CUB dataset, and a held-out text description, and draw samples conditioned on this information.

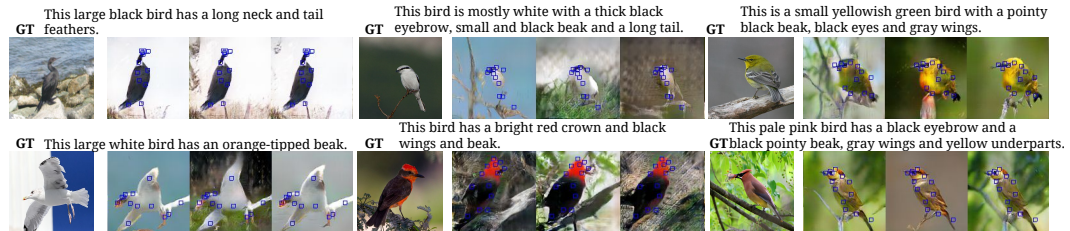

Figure 5: Bird generation conditioned on fixed groundtruth keypoints (overlaid in blue) and previously unseen text. Each sample uses a different random noise vector.

Figure 5 shows several image samples that accurately reflect the text and keypoint constraints. More examples including success and failure are included in the supplement. We observe that the bird pose respects the keypoints and is invariant across the samples. The background and other small details, such as thickness of the tree branch or the background color palette do change with the noise.

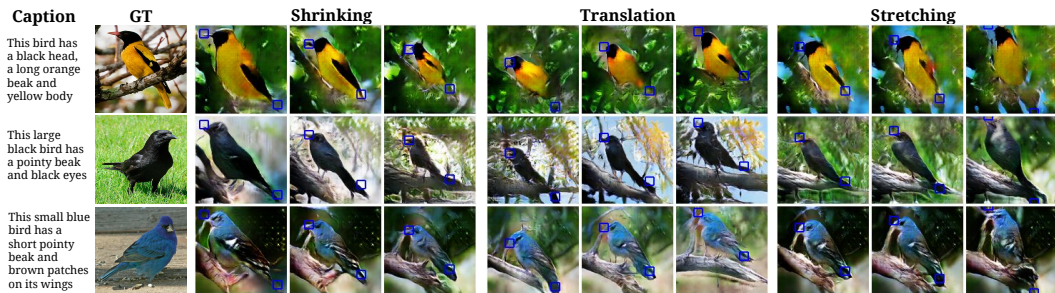

Figure 6: Controlling the bird's position using keypoint coordinates. Here we only interpolated the beak and tail positions, and sampled the rest conditioned on these two.

The GAWWN model can also use keypoints to shrink, translate and stretch objects, as shown in Figure 6. We chose to specify beak and tail positions, because in most cases these define an approximate bounding box around the bird.

Unlike in the case of bounding boxes, we can now control which way the bird is pointing; note that here all birds face left, whereas when we use bounding boxes (Figure 4) the orientation is random. Elements of the scene, even outside of the controllable location, adjust in order to be coherent with the bird's position in each frame although in each set of three frames we use the same noise vector $z$.

## 5.3 Generating both bird keypoints and images from text alone

Although ground truth keypoint locations lead to visually plausible results as shown in the previous sections, the keypoints are costly to obtain. In Figure 7, we provide examples of accurate samples using generated keypoints. Compared to ground-truth keypoints, on average we did not observe degradation in quality. More examples for each regime are provided in the supplement.

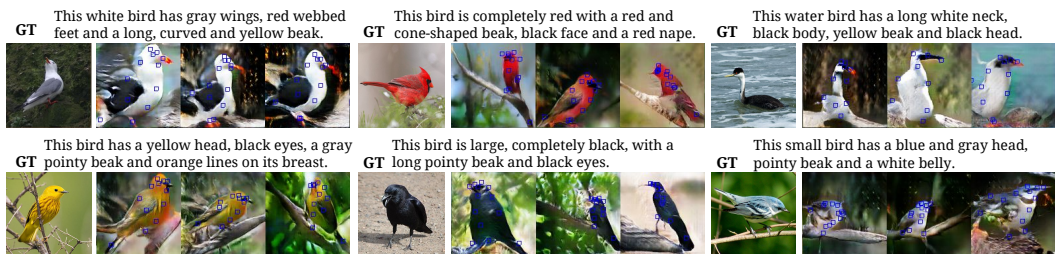

Figure 7: Keypoint- and text-conditional bird generation in which the keypoints are generated conditioned on unseen text. The small blue boxes indicate the generated keypoint locations.

## 5.4 Comparison to previous work

In this section we compare our results with previous text-to-image results on CUB. In Figure 8 we show several representative examples that we cropped from the supplementary material of [Reed et al., 2016b]. We compare against the actual ground-truth and several variants of GAWWN. We observe that the $64 \times 64$ samples from [Reed et al., 2016b] mostly reflect the text description, but in some cases lack clearly defined parts such as a beak. When the keypoints are zeroed during training, our GAWWN architecture actually fails to generate any plausible images. This suggests that providing additional conditioning variables in the form of location constraints is helpful for learning to generate high-resolution images. Overall, the sharpest and most accurate results can be seen in the $128 \times 128$ samples from our GAWWN with real or synthetic keypoints (bottom two rows).

## 5.5 Beyond birds: generating images of humans

Here we apply our model to generating images of humans conditioned on a description of their appearance and activity, and also on their approximate pose. This is a much more challenging task than generating images of birds due to the larger variety of scenes and pose configurations.

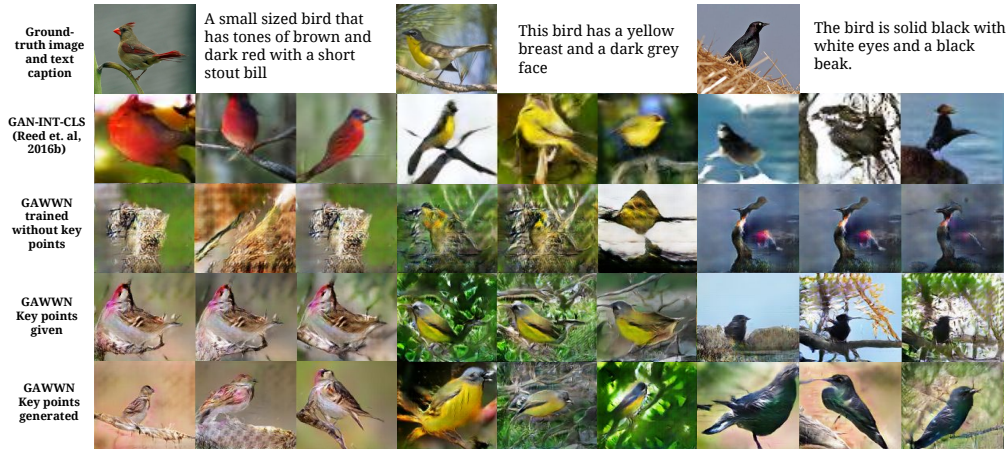

Figure 8: Comparison of GAWWN to GAN-INT-CLS from Reed et al. [2016b] and also the ground-truth images. For the ground-truth row, the first entry corresonds directly to the caption, and the second two entries are sampled from the same species.

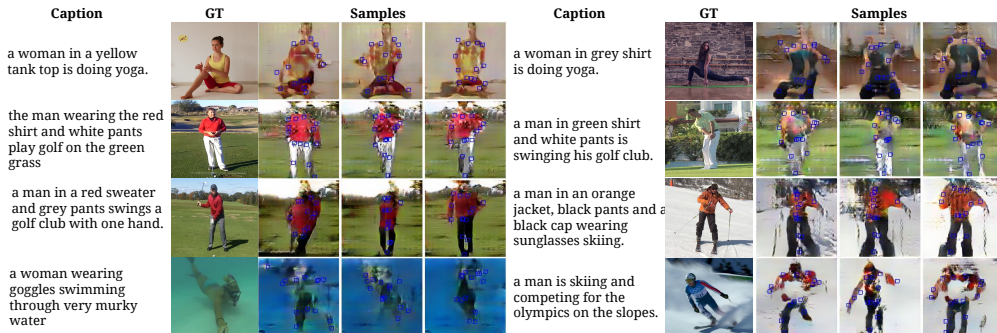

Figure 9: Generating humans. Both the keypoints and the image are generated from unseen text.

The human image samples shown in Figure 9 tend to be much blurrier compared to the bird images, but in many cases bear a clear resemblance to the text query and the pose constraints. Simple captions involving skiing, golf and yoga tend to work, but complex descriptions and unusual poses (e.g. upside-down person on a trampoline) remain especially challenging. We also generate videos by (1) extracting pose keypoints from a pre-trained pose estimator from several YouTube clips, and (2) combining these keypoint trajectories with a text query, fixing the noise vector $z$ over time and concatenating the samples (see supplement).

## 6  Discussion

In this work we showed how to generate images conditioned on both informal text descriptions and object locations. Locations can be accurately controlled by either bounding box or a set of part keypoints. On CUB, the addition of a location constraint allowed us to accurately generate compelling $128 \times 128$ images, whereas previous models could only generate $64 \times 64$. Furthermore, this location conditioning does not constrain us during test time, because we can also learn a text-conditional generative model of part locations, and simply generate them at test time.

An important lesson here is that decomposing the problem into easier subproblems can help generate realistic high-resolution images. In addition to making the overall text to image pipeline easier to train with a GAN, it also yields additional ways to control image synthesis. In future work, it may be promising to learn the object or part locations in an unsupervised or weakly supervised way. In addition, we show the first text-to-human image synthesis results, but performance on this task is clearly far from saturated and further architectural advances will be required to solve it.

**Acknowledgements**   This work was supported in part by NSF CAREER IIS-1453651, ONR N00014-13-1-0762, and a Sloan Research Fellowship.

## Footnotes

[2] Both $\phi$ and $\varphi$ could be trained jointly with the GAN, but pre-training allows us to use the best available image features from higher resolution images ($224 \times 224$) and speeds up GAN training.

[3]For details of how to apply this warping see equation 3 in [Jaderberg et al., 2015]

[4]https://github.com/soumith/dcgan.torch

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
