[Reviews · NeurIPS 2016]

Reviewer 1

Summary

The paper describes a method for generating images that allows one to specify pose and other detail from a sentence description of the object. The method is based on using a Generative Adversarial Network to train the generative model. Examples are provided of images generated from the model, showing that it is capable of generating an image from a sentence and with control over its position and size.

Qualitative Assessment

The paper builds on previous work on GAN's and sentence embeddings. The results are interesting to look at, but this seems to be more of an incremental twist on previous work rather than a fundamental advance. I don't see that it really advances our understanding of these methods. Also many of the images generated are rather garbled looking. It is difficult to know what to make of this because there is very little understanding of how the generative network actually works - i.e., what are the internal mechanisms by which it is generating images? Yes, its a neural net, but what representations does it use, and how exactly does it manipulate them to generate images? This would seem to me a more interesting question to answer.

Confidence in this Review

1-Less confident (might not have understood significant parts)


Reviewer 2

Summary

This paper describes a text-to-image generation method that is conditioned on geometric elements such as bounding and key points. It is implemented thought a local/global neural architecture that uses Adversarial Networks, a generator network and a discriminator network. The paper is clearly written and interesting on an engineering point of view because it makes sense to use keypoints and bounding boxes as an intermediary step before generating the whole image, and it might address a practical problem for designers who want to constrain the location of the generated object. Overall, the novelty of the the work is interesting, but it might not be strong or generic enough for the broad audience at NIPS. The NLP aspect is limited, so I would recommend this paper to be submitted to a vision-specialized conference such as CVPR.

Qualitative Assessment

My judgement is based on the following issues: - The choice of an adversarial Network seems a little bit arbitrary for me. Variational Auto-Encoder which is (for me at least) much simpler, more intuitive, and giving similar visual performances. The main criticism I have with adversarial networks is that both discriminative and generative networks need to be very accurate to obtain good results, while in the variational auto-encoder, we can have very good visual outputs, even if the inference network is overfitting or undercutting (of course the generative network need to be good). This is not really discussed in the existing literature on adversarial networks. There is no clear intuition why the authors choose adversarial networks. - Also, I like the idea of “what-where” network, but for me, this work could have been presented independently to the choice of adversarial networks versus variational auto-encoder. - During training, lots of images have not been labeled or have not been annotated with keypoints. The issue of incomplete information is not addressed in this work. - The approximation to the 0-1 loss is critical and relating to Akata et al. [2015] is not sufficient. Expressing directly the surrogate loss in Equation (1) would have been much clearer than a non-differentiable loss. - I'm not working in image synthesis, but at first sight, the quality of the generated images is really far from what people might expect, even for the bird dataset, we can clearly see that the image has been generated. This is really far from SIGGRAPH standards. I did not perceive the authors to be familiar with the latest data-driven techniques of photo-realistic rendering using low-level image statistics.

Confidence in this Review

2-Confident (read it all; understood it all reasonably well)


Reviewer 3

Summary

The core idea is to take advantage of labeled data which associates images with captions as well as keypoint and bounding box information in order to train conditional GANs which generate an image given the other information. The other important idea is to train another GAN to generate the missing conditioning variables, given any subset of the others (at least this was done for the keypoints, but I presume that the same principle could be applied elsewhere). A sophisticated convolutional architecture is proposed in both the generator and discriminator parts in order to take advantage of the special meaning of the conditioning variables (bounding box and keypoints).

Qualitative Assessment

This is a very interesting paper with really impressive results! One thing I would have liked to see more is a "deconstruction" of the different elements in the proposed approach, to see which ingredients matter more. Of course, part of the difficulty in such an exercise is the current lack of a quantitative evaluation procedure for GANs (or other likelihood-free generative models). Something that concerns me is the complexity of the architecture and training procedure used in these experiments. As a condition for accepting this paper, I would like the authors to confirm that they will post their code for running these experiments. Otherwise, I feel that it would be very difficult to reproduce the experiments (or a long appendix would need to be added to clarify all the experimental details). I will raise my "clarity" rating if the authors confirm that they will publish their code. For the same reason, I would expect that the modified dataset used for the MHP experiment be released publically. There appear to be strange "repeated patterns" or textures showing up in some of the generated images, like in fig 8 and fig 4. These seem to be artifacts and do not correspond to the training distribution. Do you have any clue about what is going on? Minor comments and requested clarifications: eqn between lines 99 and 100: this is not the objective typically used to actually train GANs (please clarify, mention the actual objective). sec. 3.2: why not optimize directly \phi and \Phi wrt the GAN objective? Is there any fine-tuning of these functions with the GAN objective? If so, does it make a difference? Fig 2 and 3. Clarify what non-linearities are used in hidden layers and in output layers. I am assuming rectifiers for the former and tanh for the latter, but please clarify. line 152: how was the text embedding t obtained? Like in sec 4.1? line 165: I suspect that the T should be a K. line 178: "a denoising autoencoder over keypoints would not suffice" --> actually this is probably not true, since one could run a Markov chain associated with the denoising autoencoder in order to sample from it (see "Generalized denoising auto-encoders as generative models" NIPS 2013). line 189: how are (x,y) set in k for unobserved keypoints? line 210: is this pre-training the same as for CUB? line 214: why keep the # of averaged captions always the same at training time and variable at test time? I would vary that # at training time too. line 232+: typo "an" should be "and"

Confidence in this Review

3-Expert (read the paper in detail, know the area, quite certain of my opinion)


Reviewer 4

Summary

The paper proposed an extension model of GAN, incorporating the content and location information to synthesize bird and human images.

Qualitative Assessment

- Claiming exploratory results as a contribution is bit weak in the paper, plus it is not a new dataset. The work Reed et al in ICML2016b has employed MS-COCO for the evaluation. - The work is a modification of Reed et al CVPR2016a (char-CNN-RNN) with the used of the char-CNN-GRU, but it is not show in the experiment using the char-CNN-RNN as baseline. -Page 5, line 210, is that means the authors are following the settings in the Reed et al 2016a's paper, into the proposed char-CNN-GRU ?? and the char-CNN-GPU is pre-trained on ?

Confidence in this Review

2-Confident (read it all; understood it all reasonably well)


Reviewer 5

Summary

This paper presents an interesting framework that learns to synthesize images based on unseen text descriptions. A key feature of the proposed framework is the synthesis of the object (i.e. what) and draw the object conditioned on the position/keypoint (i.e. where). The authors tested scenarios where keypoints are pre-specified, and also the scenario where keypoints are automatically generated from text descriptions. The authors then extended the experiments to generate images of humans doing various activities based on text descriptions to demonstrate the efficacy of the proposed framework.

Qualitative Assessment

The work done here is very interesting. Lots of recent works are referenced and the authors clearly highlighted the advantages of the proposed framework over other related works. In section 4.1, the authors mentioned that the processing stages are divided into the local and global stages. The technical descriptions are provided (e.g. "stride-2 deconvolutions increase spatial dimensions from 1x1 to MxM"; "Masking operation is applied [...] object bounding box are set to 0" etc.). However, it is still unclear to me why both local and global processing pathways are needed, or at least I might have missed a more detailed justification. At the very least, I would like to have obtain a clearer explanation on what exactly "local" does and what exactly "global" does, and what would happen if either one of them is absent from the processing pathway. Section 5.3: Generating both bird keypoints and images from text alone. If training the model with bird images, wouldn't the keypoints mostly fall around the same region since every training images are images of birds? I expect there'll be a much larger challenge if the training images span over different objects rather than just birds. The synthesized images look promising, especially for the bird images. The synthesized human images are acceptable as preliminary results. However, aside from image synthesis, I would also love to see a few more sentences on how the authors plan to extend and upscale this framework/application into other domains and how can other people benefit from this work. A few minor comments: - Typo in first paragraph (line 232) of Section 5.1: "[...] fixing the text an noise conditioning variables" - Typo in line 247: "Color pallete" should be "Color palette" - Typo in line 259: "degredation" should be "degradation"

Confidence in this Review

2-Confident (read it all; understood it all reasonably well)


Reviewer 6

Summary

The author(s) proposed a new model that generates images with a given description of the image content (caption) and location of the object (bounding box) or object parts (key points). They demonstrated 128 × 128 image synthesis on the Caltech-UCSD Birds dataset and the more challenging MPII Human Pose dataset. Since key points are costly to obtain, the authors suggested a text-conditional generative model of part locations which they can use at train and test time.

Qualitative Assessment

The paper suggests a new solution to an interesting problem by decomposing the problem into easier sub-problems. This resulted in an interesting image generator that controls the location and position of the objects it draws. I have two main concerns: 1. I wasn't convinced that the suggested model generates much more realistic images than previous works (Read et. al., 2016) 2. I didn't understand the intuition behind each model. Bounding-box-conditional text-to-image model: why is the text embedding replicated spatially? Keypoint-conditional text-to-image model: why were depth concatenation and point-wise multiplication both used? Conditional keypoint generation model: why is there a need to use z to estimate the location of the unobserved object parts? Moreover there are several typos: line 102: and p_g converges to p_data line 206: activitiy line 232: an noise line 267: conditiong line 289: subprobems

Confidence in this Review

2-Confident (read it all; understood it all reasonably well)